# High-Throughput Sequencing Reveals Regional Diversification of Cucurbit-Infecting Begomoviruses in Eastern Saudi Arabia

**DOI:** 10.3390/v18010075

**Published:** 2026-01-05

**Authors:** Muhammad Naeem Sattar, Sallah A. Al Hashedi, Mostafa I. Almaghasla, Sherif M. El-Ganainy, Adil A. Al-Shoaibi, Muhammad Munir

**Affiliations:** 1Central Laboratories, King Faisal University, P.O. Box 420, Al-Ahsa 31982, Saudi Arabia; sahmad@kfu.edu.sa (S.A.A.H.); adshoaibi@kfu.edu.sa (A.A.A.-S.); 2Department of Arid Land Agriculture, College of Agricultural & Food Sciences, King Faisal University, P.O. Box 420, Al-Ahsa 31982, Saudi Arabia; malmghaslah@kfu.edu.sa (M.I.A.); salganainy@kfu.edu.sa (S.M.E.-G.); 3Pests and Plant Diseases Unit, College of Agriculture and Food Sciences, King Faisal University, P.O. Box 420, Al-Ahsa 31982, Saudi Arabia; 4Department of Physics, College of Science, King Faisal University, P.O. Box 400, Al Ahsa 31982, Saudi Arabia; 5Date Palm Research Center of Excellence, King Faisal University, P.O. Box 420, Al-Ahsa 31982, Saudi Arabia

**Keywords:** begomoviruses, zucchini, snake gourd, illumina MiSeq, mixed infection, genetic diversity, Saudi Arabia

## Abstract

In Saudi Arabia, cucurbit crops such as zucchini (*Cucurbita pepo*) and snake gourd (*Trichosanthes cucumerina*) are major vegetables and key dietary components, yet their associated viral threats remain poorly understood. We surveyed symptomatic cucurbit samples from greenhouses and open fields in the Al-Ahsa and Qatif regions. The detection methods employed included PCR, RCA, and Illumina NGS. Based on nucleotide sequence comparisons and maximum-likelihood phylogenetic analysis, we identified three viruses, i.e., TYLCV, WmCSV, and ToLCPalV, present as both single and mixed infections. Sequence analyses revealed a novel strain, TYLCV-Hasa, representing a distinct lineage of TYLCV. Analysis revealed that recombination occurred solely in the DNA-A components of the identified viruses, while DNA-B segments showed no evidence of recombination. Notably, no DNA satellites were detected, suggesting cucurbits may act as independent reservoirs of begomovirus diversity. These results provide a comprehensive genomic insight into cucurbit-infecting begomoviruses in Eastern Saudi Arabia. The discovery of TYLCV-Hasa and evidence of recombination raise concerns about the emergence of novel viral variants that could pose risks to cucurbit cultivation. The results establish a foundation for advanced molecular surveillance and breeding strategies, contributing to improved food security and supporting Saudi Arabia’s Vision 2030 goals for sustainable agriculture.

## 1. Introduction

Members of the family Geminiviridae form the most extensive and diverse assemblage of ssDNA plant viruses, characterized by their distinctive geminate (twin) icosahedral particles enclosing circular single-stranded DNA genomes. These viruses pose a significant threat to agriculture in tropical and subtropical regions due to their broad host range, which spans both monocotyledonous and dicotyledonous crops, leading to substantial yield losses [1,2]. The family currently comprises 14 recognized genera. Begomovirus is the most species-rich genus, containing over 500 classified species, which are phylogenetically separated into Old World (OW) and New World (NW) evolutionary lineages [3]. Begomoviruses typically have one of two genome types: a bipartite structure consisting of two components, DNA-A and DNA-B (each approximately 2.6 kb), or a monopartite structure containing a single DNA-A component (about 2.7–2.8 kb). Their genome organization is strongly influenced by geography, with bipartite begomoviruses prevalent in the NW, while monopartite forms dominate the OW [4]. The DNA-A component of Old World begomoviruses is known to encode a basic set of six ORFs [5]. Specifically, the virion-sense strand expresses CP and AV2/V2, while the complementary strand expresses Rep, TrAP, REn, and C4. Notably, some recent studies on monopartite species have reported the existence of extra ORFs like V3 and C5–C7 [6,7]. Meanwhile, DNA-B encodes two key proteins: BV1, the nuclear shuttle protein (NSP), and BC1, the movement protein (MP), both of which are essential for viral transport within and between host cells [8]. Both DNA components share a ~200-nucleotide common region (CR) containing the replication origin and a conserved nonanucleotide motif (TAATATTAC), both of which are crucial for both replication and bidirectional transcription. In the OW, it is common for monopartite begomoviruses to be found in association with ssDNA-satellites, i.e., betasatellites, alphasatellites, and deltasatellites, which are known to influence viral pathogenicity [9,10]. For example, betasatellite-encoded βC1 is widely recognized as the primary contributor to symptom development [11]. However, a recently described protein βV1 also enhances virulence [12]. Moreover, the interaction of the virus with alphasatellites and deltasatellites introduces additional layers of complexity to the disease mechanism and the plant’s host response [13].

The global dissemination of begomoviruses is driven by the polyphagous whitefly vector (*Bemisia tabaci*), the trade in infected plant material, the viruses’ high adaptability through recombination, and mixed infections [14,15]. Although local begomovirus populations often retain genetic distinctiveness, their evolution can occur through localized adaptation or parallel diversification [16]. The transboundary dissemination of these viruses is an increasingly documented phenomenon. For instance, Tomato yellow leaf curl virus (TYLCV) has progressively expanded westward into Africa [17], while Squash leaf curl virus has been detected throughout the Middle East [18]. Similarly, cotton leaf curl disease has shown an eastward spread reaching China [19]. Moreover, Watermelon chlorotic stunt virus (WmCSV), originally confined to the Old World, has now been reported in both the United States and Mexico [20,21]. Another closely related virus, tomato leaf curl Palampur virus (ToLCPalV), first described in India, has subsequently been identified across several Middle Eastern countries, including Iran, Iraq, Oman, and Saudi Arabia [22,23,24]. Several regional studies also highlighted distinct evolutionary patterns for TYLCV in the Middle East. Hosseinzadeh, et al. [25] proposed Iran as a center of TYLCV diversification, reinforced by a recent genetic diversity analysis by Akbar, et al. [26], suggesting that many TYLCV variants circulating in the Arabian Peninsula likely originated from, or are closely linked to, Iranian lineages.

Both monopartite and bipartite begomoviruses are documented in Saudi Arabia. Key bipartite species include WmCSV, ToLCPalV, and tomato leaf curl Sudan virus (ToLCSDV). The main monopartite begomoviruses impacting crops are TYLCV and cotton leaf curl Gezira virus (CLCuGeV) [27,28,29,30]. Notably, ToLCPalV has been recently reported to cause mixed infections in tomato and cucurbit crops within the Al Ahsa region [22,31].

Among susceptible hosts, zucchini (*Cucurbita pepo* L.) and snake gourd (*Trichosanthes cucumerina* L.) are widely grown cucurbits with high economic importance. Globally, zucchini contributes to approximately 23.4 million tonnes of annual squash production (https://www.futuremarketinsights.com/reports/squash-market; accessed on 15 May 2025). In Saudi Arabia, zucchini cultivation has expanded under greenhouse-based systems, with total vegetable output in regions like Qassim exceeding 115,000 tonnes annually [32]. Although less commercially prominent than zucchini, snake gourd is also cultivated across tropical Asia and the Middle East and remains a key dietary and regional cucurbit crop [33]. Both hosts are vulnerable to begomoviruses, which cause characteristic symptoms like chlorotic mosaic, interveinal yellowing, curling, vein thickening, and stunting, ultimately reducing yield and marketable quality, especially under high whitefly infestations [28,30]. While the number of begomovirus reports in the Arabian Peninsula is increasing, the full extent of the genetic diversity of the circulating viral genomes remains poorly understood because molecular surveillance has primarily relied on traditional, primer-dependent methods [27,34]. These approaches are often constrained by primer selectivity and the dominance of specific viral populations, making it difficult to accurately assess the viral diversity in infected hosts. Although cucurbits have substantial agricultural importance, their viral pathogens in Saudi Arabia remain poorly understood, and the identity, diversity, and genome organization of begomoviruses infecting zucchini and snake gourd remain largely uncharacterized. This highlights the need for a comprehensive investigation of begomovirus infections in cucurbits using next-generation sequencing (NGS).

In this study, we aimed to detect, identify, and molecularly characterize begomovirus strains infecting zucchini and snake gourd in the Eastern regions of Saudi Arabia using an NGS approach. Field surveys in Al-Ahsa revealed symptomatic plants from both open fields and greenhouse environments. While preliminary PCR diagnostics confirmed the presence of begomovirus, and NGS revealed mixed infections involving WmCSV, TYLCV, and ToLCPalV, occurring as either single or co-infections. These findings highlight the complex viral landscape affecting cucurbit production in the region. By integrating PCR-based diagnostics, rolling circle amplification (RCA), NGS, and phylogenetic analysis, this study provides new insights into begomovirus diversity in zucchini and snake gourd and establishes a basis for improved disease surveillance and management strategies in Saudi agriculture.

## 2. Materials and Methods

### 2.1. Collection of Plant Samples and Preliminary Begomovirus Screening

Leaf tissues exhibiting characteristic begomovirus-associated symptoms, including interveinal yellow mottle/mosaic, vein banding or clearing, upward leaf curling, and deformation of young foliage with reduced leaf size and mild stunting (Figure 1A, zucchini; B, snake gourd), were collected from 13 zucchini and six snake gourd plants. The samples were gathered from eight different greenhouse and field plots in the Al-Hofuf and Qatif regions of Saudi Arabia (Appendix A). Immediately after collection, samples were snap-frozen in liquid nitrogen and subsequently held at −80 °C until DNA extraction.

Genomic DNA was extracted using a commercial silica-based kit (Qiagen, Redwood City, CA, USA) following the manufacturer’s protocol. Initial virus screening was conducted through PCR using the universal degenerate primers AC1048/AV494 [35], which are designed to target a ~550 bp region of the begomoviral CP gene. Subsequently, the amplified DNA fragments were purified with the GeneJet PCR Purification Kit (ThermoFisher Scientific, Waltham, MA, USA) and were sequenced bidirectionally at the Macrogen facility in Korea.

### 2.2. Full-Length Genome Recovery by Rolling Circle Amplification and Next-Generation Sequencing

To obtain complete begomovirus genome sequences, RCA was performed on DNA extracted from all positive samples using Φ29 DNA polymerase (GE Healthcare, Chicago, IL, USA). RCA-enriched products were purified from four zucchini and two snake gourd samples and processed for NGS (Table 1). For six representative samples, libraries were prepared with Nextera XT and sequenced as paired-end reads (2 × 300 bp) on an Illumina MiSeq platform at Macrogen (Seoul, Republic of Korea). This approach is designed to capture circular DNA viruses but does not detect RNA viruses. The protocol used was consistent with previously published reports [22,31].

### 2.3. Genome Assembly and Bioinformatic Analysis of NGS Data

Raw reads generated in FASTQ format were initially evaluated for quality using FastQC v0.11.8 [36]. Raw reads were quality-filtered with Trimmomatic v 0.39 to remove adapters and low-quality bases, and the post-trim metrics were inspected in FastQC to verify improvement [37]. We used BWA-MEM2 v2.2.1 [38] to map the cleaned sequence data to a set of begomovirus reference genomes, including TYLCV, and the respective DNA-A and DNA-B segments for both ToLCPalV and WmCSV. Resulting SAM files were converted, sorted, and indexed as BAM with SAMtools v1.9, and Picard was used to harmonize read-group annotations. Consensus genomes were derived with SAMtools mpileup and iVar v1.3 [39], applying a minimum depth cutoff of 20× for base acceptance.

### 2.4. PCR Validation of Viral Genome Components

To verify the viral sequences derived from NGS, RCA products from infected samples were diluted (1:10) and re-amplified thereafter, using component-specific primers as described previously [31]. The resulting amplicons were purified and subjected to a bidirectional Sanger sequencing approach at Macrogen Inc. (Seoul, Republic of Korea). Sequence identities were established by conducting BLASTn searches against the NCBI GenBank database (https://blast.ncbi.nlm.nih.gov/Blast.cgi; accessed on 1 April 2025).

### 2.5. Sequence Alignments and Identity Analysis

The pairwise nucleotide (nt) identities were computed for all query sequences, after which the most similar reference genomes were retrieved from GenBank for comparative evaluation. For each component (DNA-A and DNA-B), we built full-length multiple-sequence alignments with ClustalW in MEGA12 [40]. We subsequently calculated pairwise nt sequence identity percentages using the Sequence Demarcation Tool (SDT v1.2), adhering to the standard classification guidelines for geminiviruses [41]. Coding and intergenic regions were annotated with the NCBI ORFfinder tool (available at https://www.ncbi.nlm.nih.gov/orffinder/, accessed on 15 May 2025).

### 2.6. Phylogenetic Analysis

Phylogenetic relationships among begomovirus genomic components were reconstructed by maximum likelihood (ML) in MEGA v12. Model testing indicated that the General Time Reversible model with a gamma-distributed rate variation (GTR+I) best described the data, and this model was applied in subsequent analyses. Resulting trees were exported in Newick format and graphically refined using iTOL v6.5 and Adobe Illustrator CC 2025.

### 2.7. Detection of Recombination Events

To investigate recombination patterns, a dataset was compiled consisting of 100 complete begomovirus DNA-A and 100 DNA-B genomic component sequences, including those obtained in this study. Only sequences sharing >70% nt identity with the study isolates were included, following the criteria of Crespo-Bellido, et al. [42]. The datasets were exported as FASTA files after being aligned in MEGA12. Recombination events were identified using two programs, GARD and RDP5 v5.0 [43]. We utilized the seven algorithms contained in the RDP5 suite and enforced a strict validation criterion: only events supported by at least three separate methods were accepted. A Bonferroni-adjusted *p*-value of 0.05 was applied to assess significance, and default settings were used for all program runs.

### 2.8. Genetic Diversity, Neutrality, and Population Differentiation Analyses

To complement the genomic characterization of cucurbit-associated begomoviruses obtained through high-throughput sequencing, we performed a targeted regional population-genetic analysis. This analysis aimed to assess whether Iranian (IR) and Arabian-Red Sea (AR) begomovirus populations exhibit measurable differences in nt diversity and demographic signatures. This analysis focused specifically on WmCSV and ToLCPalV because (i) these viruses were prevalent in cucurbits in Eastern Saudi Arabia, and (ii) detailed population-genetic studies already exist for TYLCV in Iran and the Arabian Peninsula [25,26]. Therefore, TYLCV was not re-evaluated in this study.

#### 2.8.1. Dataset Definition and Population Delineation

Full-length DNA-A and DNA-B sequences of ToLCPalV and WmCSV were retrieved from GenBank and supplemented with Saudi Arabian sequences generated in this study. To avoid sampling biases from regions with disproportionate sequencing efforts and to maintain a focus on cross-border viral movement relevant to Eastern Saudi Arabia, the dataset was restricted to two geographically defined populations, i.e., Pop-1 (IR): Iran; and Pop-2 (AR): Arabian-Red Sea corridor (Saudi Arabia, Oman, Iraq, Yemen, Kuwait, Sudan). We acknowledge that the Pop-1 dataset comprises fewer sequences and represents historical data compared to the contemporary and larger Pop-2 dataset. While this disparity is a common limitation in retrospective analyses, the comparison provides valuable insights into potential demographic shifts over time.

Partial genomes, sequences containing >3 ambiguous nts, redundant duplicates, or recombinant fragments supported by ≥3 RDP5 algorithms (Bonferroni-corrected *p* ≤ 0.05) were removed from the dataset. Separate alignments were constructed for each virus and genome component (DNA-A and DNA-B).

#### 2.8.2. Multiple Sequence Alignment and Recombination Screening

The sequences were aligned in MEGA12 using the MUSCLE algorithm and manually inspected to maintain codon structure. The best-fit nt substitution model for each dataset was selected according to the Bayesian Information Criterion (BIC). Recombination screening was performed using RDP, GENECONV, BootScan, MaxChi, Chimaera, SiScan, and 3Seq under default parameters. Recombinant sequences or breakpoints identified by at least three methods were removed from downstream analysis to prevent inflation of diversity statistics.

#### 2.8.3. Genetic Diversity and Neutrality Tests

Genetic diversity indices were estimated in DnaSP v6.12, including the number of polymorphic sites (S), total mutations (Eta), number of haplotypes (h), haplotype diversity (Hd), nt diversity (π), average nucleotide differences (k), and Watterson’s estimator (θw). Neutrality tests, including Tajima’s D (TD) and Fu & Li’s D (FLD), were calculated using 10,000 coalescent simulations to test for departures from mutation-drift equilibrium. Gaps and missing sites were treated by pairwise deletion. Analyses were performed separately for DNA-A and DNA-B components of both viral species.

#### 2.8.4. Population Differentiation and Gene Flow Statistics

To assess the degree of genetic differentiation between Pop-1 and Pop-2, we calculated differentiation indices, including K_ST_ and the nearest-neighbor statistics (S_nn_), using DNASP v6.12. Additionally, the pairwise fixation index (F_ST_) and Analysis of Molecular Variance (AMOVA) were performed using Arlequin v3.5.2.2. The significance of the F_ST_ values was determined using a non-parametric permutation test with 1000 replicates. AMOVA was used to partition the total molecular variance into components attributable to differences among populations and within populations.

## 3. Results

### 3.1. Next-Generation Sequencing and Data Analysis

During a survey of eight greenhouse and field plots in the Al-Ahsa region (including Al-Hofuf and Qatif municipalities), Saudi Arabia, we observed that approximately 40–60% of zucchini plants and 20–30% of snake gourd plants displayed characteristic begomovirus-like symptoms with associated whitefly infestations. Symptoms in zucchini were severe, including bright yellow mosaic/mottle with vein-clearing and banding, laminar rugosity/blistering, upward cupping and curling, apical leaf clustering, and a marked reduction in leaf size and plant stature (Figure 1A). In snake gourd, symptoms were milder, predominating on young leaves as interveinal yellow mosaic patches with marginal curling/puckering and limited stunting (Figure 1B). These profiles are consistent with typical begomovirus infections in cucurbit hosts.

We confirmed begomovirus infection in 12 zucchini and two snake gourd samples collected from Al-Hofuf and Qatif using primers targeting the core CP gene. Subsequent Sanger sequencing of the amplicons revealed strong sequence identity with the ToLCOMV and WmCSV DNA-A components. Based on these initial findings, four zucchini samples (SqH1, SqH42, SqSK7, and SqA2) and two snake gourd samples (SqH33 and SqA8my) were selected for NGS analysis using the Illumina high-throughput sequencing platform. From these six samples, we generated six DNA libraries, producing 2,380,606, 2,299,758, 2,567,202, 2,262,856, 2,728,541, and 2,422,677 raw paired reads per sample, which were assembled into full-length begomovirus genomes, including both bipartite and monopartite species. The sequences recovered during this study have been deposited into the NCBI GenBank database, each receiving a unique accession number (Table 1). Notably, no DNA-satellites were detected in any of the samples analyzed.

### 3.2. Deciphering Begomovirus Genomes Through Comparative Sequence Analysis

Pairwise sequence identity analysis using SDT demonstrated that most of the samples were infected with a single begomovirus species. This included the zucchini samples (SqH1, SqH41, and SqSK7) and the snake gourd samples (SqH33 and SqA8my). However, the zucchini sample SqA2 from the Qatif region was distinctive in containing a mixed infection of begomoviruses. An examination of the full-length sequences SHT1, SHT2, and SQT1 showed mutual identity ranging from 93.8 to 98.8% (Figure 2A). When compared to GenBank entries, SHT1 (from SqH1 zucchini) and SQT1 (from SqSK7 zucchini) showed nt sequence identities in the range of 98.0 to 99.7% with three existing TYLCV isolates from Saudi Arabia, which originated from cucumber (OR865128) and Malva (OL416210 and OL416216). In contrast, these two sequences showed less than 94% nt identity with other TYLCV isolates. Finally, the sequence SHT2 from the snake gourd sample (SqH33) showed it was most closely related (95.8% nt sequence identity) to the “Boushehr” strain isolate (GU076454) from Iran [44], as shown in Figure 2A. In the ML phylogenetic analysis of the full-length begomovirus sequences, the three isolates from this study showed distinct evolutionary relationships (Figure 3A). The isolate SHT2 clustered most closely with the Iranian “Boushehr” strain of TYLCV (GU076454), along with other TYLCV isolates from Sudan and Iran, suggesting an introduction from outside the region. In contrast, isolates SHT1 and SQT1 grouped with previously reported Saudi isolates, indicating local circulation and possible in-country diversification. Based on nt sequence comparisons and the ICTV criteria for begomovirus species demarcation (>92%) and strain demarcation (>94%) [45], we propose a strain-level group, TYLCV-Hasa, comprising SHT1, SQT1, and three previously reported Saudi TYLCV genomes (OR865128, OL416210, and OL416216). We denote the focal genome as TYLCV-Hasa [SA:Zuc:18].

The sequences SHWA1, SQWA1, and SQWA2 shared mutual nt sequence identities of 94.3–99.8%. The sequences SHWA1 from the zucchini sample (SqH41) and SQWA1 from the snake gourd sample (SqA8my) were 98.7% and 98.5% identical, respectively, to the WmCSV isolate (PP320240) from snake gourd in Saudi Arabia [30]. The SQWA2 sequence, obtained from the zucchini sample (SqA2), exhibited a maximum nt sequence identity of 96.0% with WmCSV isolates previously reported from both cucumber and snake gourd in Saudi Arabia [31]. In the resulting phylogenetic dendrogram, these three isolates were placed within the WmCSV clade, showing close relationships to other Saudi Arabian WmCSV isolates and further supporting their endemic status.

The sequences SHWB1, SQWB1, and SQWB2 demonstrated a high degree of mutual nt sequence identity, ranging from 95.7 to 100%. Specifically, SHWB1 (from zucchini sample SqH41) and SQWB1 (from snake gourd sample SqA8my) shared their highest nt identity (96.3–96.7%) with a WmCSV isolate from Iran (AJ245653) [46], as illustrated in Figure 2B. Conversely, sequence SQWB2 (from zucchini sample SqA2) showed a 98.8% nt sequence identity with a WmCSV isolate previously reported in Saudi Arabian cucumber (OR865138) [31]. Our phylogenetic analysis confirmed that these isolates belong to the WmCSV DNA-B clade, where they formed a robustly supported sub-lineage alongside other regional and Saudi Arabian isolates. The tight grouping of these isolates, supported by high bootstrap values, indicates their divergence from a recent common ancestor and highlights their localized circulation within the cucurbit-growing regions of Saudi Arabia.

The nt sequence identity of SQPA1 from the zucchini sample (SqA2) was computed as the highest (97.6%) with the ToLCPalV DNA-A isolate (OR865137) [31]. Isolate SQPA1 (PV645676) was positioned within clade-2 of the ToLCPalV clade in the phylogenetic dendrogram, grouping with sequences from both Saudi Arabia and other Middle Eastern countries, which implies its widespread regional circulation (Figure 3A). The sequence SQPB1 showed a very high (99.7%) nt sequence identity match with a previously reported DNA-B component (OR865145) of the ToLCPalV isolate identified from cucumber in Saudi Arabia [31]. In the DNA-B phylogenetic tree, isolate SQPB1 grouped within clade-2 of ToLCPalV DNA-B, showing close phylogenetic affinity with other Saudi Arabian isolates such as OR865146 and OR865145. This cluster was supported by strong bootstrap values, indicating a robust evolutionary relationship. The placement of PV645671 within clade-2, together with Middle Eastern isolates, suggests regional adaptation and potential cross-border virus movement, consistent with the dynamic epidemiology of begomoviruses in cucurbit hosts.

Detection of each begomovirus component in the corresponding zucchini and snake gourd samples was confirmed with previously reported, component-specific primer sets [31].

### 3.3. Exploring Potential Recombination Events

Recombination was first examined using the automated GARD tool (Datamonkey online source, https://www.datamonkey.org/, accessed on 20 July 2025). This analysis identified both strong and subtle recombination signals exclusively within the begomoviral DNA-A sequences (Figure 4A). Conversely, neither the GARD results nor the subsequent RDP analysis detected any significant recombination signals in the DNA-B component of the identified begomovirus genomes.

Comprehensive recombination screening with the RDP suite revealed multiple, well-supported events across the analyzed begomovirus isolates (Table 2, Figure 4). For SHT1, two independent recombination breakpoints were identified.

The first spanned nt positions 97–1958, with TYLCV (GU076446) acting as the major parental sequence and TYLCV_SQT1 as the minor contributor (*p* = 5.84 × 10^−28^). The second region, covering positions 1959 and 2112, involved TYLCV (DQ631892) and TYLCV (HG941641) as the respective major and minor parents (*p* = 2.75 × 10^−12^). For SQT1, three distinct recombination signals were detected. The first occurred between positions 1172 and 1273, derived from TYLCV (EF423426) as the major parent and WmCSV_SQWA2 as the minor sequence. A second event, between 1941 and 2093, also involved TYLCV (DQ631892) and TYLCV (HG941641) as the putative parents. A third event extended across positions 2661 and 2717, where TYLCV (KJ830842) served as the major parent and an unidentified TYLCV isolate (ON756218) as the minor parent, supported by a highly significant probability (*p* = 3.30 × 10^−12^). In SHT2, two recombination sites were identified. The first, between positions 1172 and 1273, showed TYLCV (EF423426) as the major parent and WmCSV_SQWA2 as the minor parent. The second event, spanning 2105 and 2752, was formed between TYLCV_SQT1 and an uncharacterized TYLCV isolate (JQ928348), both exhibiting strong statistical support (*p* = 7.96 × 10^−11^ and *p* = 1.15 × 10^−69^, respectively).

For SQPA1, a single recombination breakpoint was recorded between positions 1158–1241, with TYLCV (GU076446) identified as the major parent and TYLCV_SQT1 as the minor parental sequence (*p* = 5.61 × 10^−12^). Finally, SQWA2 demonstrated three independent recombination regions: one extending from 160 to 276 (between WmCSV OK058529 and TYLCV_SQT1), a second between 344 and 506 (between WmCSV KJ958912 and TYLCV_SQT1), and a third between 1718 and 1854 (between WmCSV PP320241 and TYLCV OR865126). All three events were consistently confirmed by multiple detection algorithms, with *p*-values ranging from 4.97 × 10^−16^ to 3.35 × 10^−19^.

### 3.4. Genetic Diversity Analysis of Local Begomovirus Populations

The diversity analyses showed clear differences between Pop-1 and Pop-2 for both ToLCPalV and WmCSV (Table 3, Figure 5). For ToLCPalV DNA-A, Pop-1 displayed moderate diversity (π = 0.009; S = 204), while Pop-2 showed higher diversity (π = 0.014; S = 210). Haplotype diversity was high in both groups (Hd = 0.96–1.00). Watterson’s estimator (θw) exceeded π in both populations, indicating an excess of low-frequency variants (Figure 5); this gap was largest in Pop-1 and corresponded to the strongly negative neutrality values (Tajima’s D = −2.26; Fu & Li’s D = −3.47).

A similar trend was observed for ToLCPalV DNA-B, where Pop-2 exhibited greater polymorphism (S = 379) and nt diversity (π = 0.040) compared with Pop-1 (S = 287; π = 0.026). Again, θw was higher than π in both populations (Table 3; Figure 5), consistent with the mildly negative neutrality statistics in Pop-1 and near-neutral values in Pop-2.

For WmCSV DNA-A, Pop-1 showed relatively low diversity (π = 0.010; S = 76), whereas Pop-2 displayed substantially higher variation (π = 0.019; S = 262). This pattern was most pronounced in WmCSV DNA-B, where Pop-2 had the highest diversity across all datasets (π = 0.044; S = 510). In all WmCSV datasets, θw remained greater than π, producing consistently negative but moderate neutrality values. Overall, the Arabian–Red Sea population showed higher nt diversity across both viruses, while the Iranian population exhibited stronger departures from neutrality, reflecting distinct demographic histories. It should be noted that the lower diversity observed in Pop-1 may be partially influenced by the smaller sample size and older collection dates relative to Pop-2.

### 3.5. Population Differentiation Analysis

The genetic structure of ToLCPalV and WmCSV populations was further investigated through differentiation indices and AMOVA (Table 4). All viral components exhibited statistically significant genetic differentiation between Pop-1 and Pop-2 (*p* < 0.05). The highest level of differentiation was observed in ToLCPalV DNA-A (F_ST_ = 0.178, *p* < 0.001), supported by a high S_nn_ value of 0.907 and a K_ST_ of 0.046. ToLCPalV DNA-B also showed highly significant differentiation (F_ST_ = 0.106, *p* < 0.001), with an even higher S_nn_ value of 0.931, suggesting very distinct haplotype compositions between the two regions.

For WmCSV, DNA-A showed moderate but significant differentiation (F_ST_ = 0.118, *p* = 0.006), with a high S_nn_ (0.967) indicating strong population subdivision. WmCSV DNA-B displayed the lowest differentiation among the analyzed components (F_ST_ = 0.093, *p* = 0.031), though the result remained statistically significant.

AMOVA results were consistent with the fixation indices, revealing that while most of the genetic variation exists within populations (82.25–90.67%), a significant proportion of the total variation is attributed to differences among populations (Table 4). Specifically, ToLCPalV DNA-A showed the highest variation among populations (17.75%), followed by WmcSV DNA-A (11.97%), ToLCPalV DNA-B (10.61%), and WmCSV DNA-B (9.33%).

## 4. Discussion

The global expansion and increasing incidence of begomoviruses pose significant threats to agricultural productivity [16], particularly in economically important crops like zucchini and snake gourd [28,30]. In Saudi Arabia, where cucurbit cultivation is expanding, understanding the genetic diversity and distribution of circulating begomoviruses is essential for designing effective and sustainable control measures. This study utilized an NGS approach to comprehensively characterize begomovirus strains infecting symptomatic zucchini and snake gourd plants from the Al-Ahsa region of Saudi Arabia. The results demonstrated the co-presence of TYLCV, WmCSV, and ToLCPalV, occurring either individually or in mixed infections within the same hosts.

Our results corroborate those of previous regional reports, which confirm the prevalence of both monopartite and bipartite begomoviruses in the Arabian Peninsula [22,27,28,30,31]. The observed disease incidence (40–60% in zucchini and 20–30% in snake gourd) and characteristic symptoms, such as interveinal chlorosis, yellow mosaic patterns, leaf curling, and stunting, are consistent with typical begomovirus infections in cucurbit hosts. Beyond begomoviruses, cucurbits in Saudi Arabia face significant threats from RNA viruses. Zucchini yellow mosaic virus (ZYMV), watermelon mosaic virus (WMV), and papaya ringspot virus (PRSV) are the dominant pathogens causing severe disease across the region [47,48]. In Al-Ahsa, cucumber mosaic virus (CMV) often complicates diagnosis [49], while cucurbit yellow stunting disorder virus (CYSDV) remains a persistent issue in western Saudi Arabia [50]. This complex viral landscape emphasizes the need for broad-spectrum surveillance strategies.

While begomoviruses have been reported previously in Saudi Arabia and neighboring countries, the novelty of this work lies in three key findings: (i) the identification of a distinct “Hasa” strain of TYLCV, which clusters with Saudi Arabian isolates but is divergent from other known strains; (ii) the detection of multiple recombination events involving TYLCV, WmCSV, and ToLCPalV, underscoring the dynamic nature of their evolution; and (iii) the simultaneous recovery of full-length genomes from both zucchini and snake gourd, which expands the known host range and confirms cucurbits as important reservoirs of begomovirus diversity.

Phylogenetic analysis revealed contrasting evolutionary trajectories and origins of the identified begomovirus isolates. Two TYLCV isolates (SHT1 and SQT1) grouped tightly with previously reported Saudi Arabian isolates, supporting their local diversification. We propose designating these as members of a new “Hasa” strain of TYLCV. Conversely, SHT2 clustered with the Iranian “Boushehr” strain, suggesting a regional introduction through cross-border virus movement, consistent with the role of trade and whiteflies in mediating long-distance dispersal [14,17]. The WmCSV DNA-A and DNA-B isolates (SHWA1, SQWA1, SQWA2, SHWB1, SQWB1, and SQWB2) grouped into a Saudi Arabian sub-lineage, confirming their endemic establishment and localized circulation within cucurbit-growing regions, reinforcing our previous findings [28,31]. The observed patterns are consistent with WmCSV becoming well-adapted and diversified over time, which highlights its successful and long-term presence in the region. In parallel, the ToLCPalV DNA-A and DNA-B isolates (SQPA1 and SQPB1) clustered within clade-2 alongside isolates from Saudi Arabia and the broader Middle East. This phylogenetic placement supports previous documentation regarding its wide regional dissemination [22,23,31,51]. ToLCPalV, which originated in India, has spread eastward into Pakistan and westward into the Middle East [24]. The recurrent discovery of ToLCPalV from Saudi Arabia suggests that it has been widening its distribution in this region. The detailed analysis by Sattar, et al. [24] indicated that ToLCPalV isolates from the Arabian Peninsula and the Indo-Pak subcontinent are genetically quite distinct. While they share a common ancestor, they have evolved independently, as is evident from the clustering of these isolates into two distinct clades (Figure 3A,B). Together, these results strengthen the idea that Saudi Arabia serves as a regional hub for the circulation and diversification of these begomoviruses.

Our recombination analysis provided compelling evidence of intra- and inter-species genetic exchanges. TYLCV isolates were found to have recombined with both ToLCPalV and WmCSV. Likewise, WmCSV isolate SQWA2 displayed three distinct recombination events involving various WmCSV and TYLCV isolates. Our findings highlight extensive recombination activity among Saudi Arabian cucurbit begomoviruses, driven by mixed infections and the polyphagous nature of the whitefly vector, B. tabaci. Recombination is a well-established driver of begomovirus evolution, enabling host adaptation, virulence shifts, and resistance-breaking [42,52]. The detection of recombination breakpoints within functional genomic regions underscores the likelihood of adaptive consequences for viral fitness and epidemiology. The absence of significant recombination signals in the DNA-B component, as observed in our study, is consistent with some previous reports and could be attributed to different evolutionary pressures or less frequent genetic exchange in this component [21,53]. However, the exact reasons for this differential recombination pattern warrant further investigation.

The contrasting diversity patterns between Iran and the Arabian–Red Sea region indicate distinct evolutionary histories among regional begomoviruses. For both WmCSV and ToLCPalV, the Arabian–Red Sea population showed higher nt diversity, especially in DNA-B, consistent with a genetically heterogeneous and expanding viral pool shaped by repeated introductions and extensive whitefly movement. In contrast, the Iranian population displayed stronger negative neutrality indices and larger θw–π gaps, reflecting an excess of rare variants and a more structured, long-standing viral reservoir. This pattern aligns with earlier work on TYLCV, where full-genome analyses identified greater genetic variability and recombination hotspots in southern Iranian isolates, leading Hosseinzadeh, et al. [25] to propose Iran as a center of TYLCV diversification; Akbar, et al. [26] further showed that Iranian isolates form the basal core of a monophyletic cluster shared with Oman, Saudi Arabia, Pakistan, and Kuwait, indicating historical spread from Iran into the Arabian Peninsula. Our findings also parallel the regional structure reported for ToLCPalV [24], reinforcing that Iranian populations likely represent older, more constrained lineages, whereas Arabian-Red Sea populations constitute more dynamic and admixed viral assemblages shaped by cross-border movement and agricultural connectivity. This regional population structure is statistically confirmed by our differentiation analysis (Table 4), which reveals significant F_ST_ values for all analyzed components. The relatively high F_ST_ (0.178) and among-population variation (17.75%) for ToLCPalV DNA-A suggest limited gene flow between Iranian and Arabian-Red Sea populations for this component, maintaining their distinct genetic identities. This finding aligns with recent phylogeographic analyses of ToLCPalV, which described genetically distinct lineages circulating in the Arabian Peninsula versus the Indo-Pak subcontinent [24]. In contrast, the lower differentiation observed for WmCSV DNA-B (F_ST_ = 0.093) might reflect a higher tolerance for reassortment or more recent shared ancestry. Despite these differences, the consistently high S_nn_ values (>0.80) across all datasets strongly support the conclusion that the Iranian and Arabian-Red Sea viral populations are genetically distinct subdivisions rather than a single panmictic population. These results mirror evolutionary patterns previously reported for TYLCV, where Iranian isolates formed a basal, structured population distinct from the expanding lineages found in the Arabian Peninsula [25,26].

Interestingly, no DNA-satellites were detected by our RCA-NGS pipeline and satellite-directed PCR screens. This suggests that symptoms can be attributed to begomovirus components alone or that satellites were below our detection threshold or absent from the sampled tissues/time points. While the absence of satellites in OW begomoviruses is uncommon, it has been reported in certain agroecosystems where helper viruses are sufficiently competitive [13,54]. Recently, ToLCPalV was reported to infect zucchini in association with tomato leaf curl betasatellite (ToLCB) from Iran [55], and such associations have been documented for different begomoviruses and DNA-satellites [10,56]. In another study, however, Al-Waeli, et al. [57] could not find any association of DNA-satellites with ToLCPalV in Iraq. Their absence in the sequenced samples from our study could be due to methodological factors, such as sequencing depth or RCA bias. Further studies focusing on the presence and role of these satellite molecules in the context of Saudi Arabian cucurbit begomoviruses would be valuable.

This study demonstrates the practical utility of NGS for resolving the complex begomovirus populations in agricultural ecosystems. Since our objective was to assemble high-confidence consensus genomes for taxonomic identification rather than to analyze low-frequency intra-host variants (quasispecies), technical replicates were not employed. The high depth of coverage (>20×) and strict quality filtering ensured the reliability of the dominant viral sequences reported here. By moving beyond traditional primer-dependent methods, NGS provides an unbiased and comprehensive view of viral diversity, enabling the identification of both known and potentially novel viral strains and recombination events [34,58]. It is important to note that while RCA is highly effective for enriching circular DNA for discovery, it may introduce amplification bias and alter the relative proportions of viral components [59]. Therefore, our findings represent a qualitative identification of the circulating viral diversity rather than an absolute quantification of viral load. Our findings underscore the limitations of traditional methods. For example, our previous studies using conventional techniques on snake gourd found only WmCSV infection [30], whereas NGS approaches revealed complex mixed infections involving WmCSV, TYLCV, and ToLCPalV. Our current findings are consistent with earlier work showing that NGS is a successful method for identifying mixed infections of multiple begomoviruses and DNA-satellites in crops such as tomato and muskmelon [22] and cucumber [31]. The detection of multiple begomoviruses and recombinants in zucchini and snake gourd highlights a serious risk to Saudi Arabian cucurbit production. As a high-value greenhouse crop, zucchini is particularly susceptible, with high whitefly infestations creating a conducive environment for virus spread and mixed infections. This robust molecular characterization provides a critical foundation for developing targeted and sustainable disease management strategies, including the deployment of resistant varieties, vector control measures, and improved diagnostic tools.

## 5. Conclusions

The study employed NGS to resolve a complex begomovirus landscape affecting cucurbit production in Eastern Saudi Arabia, revealing the co-circulation of TYLCV, WmCSV, and ToLCPalV in both single and mixed infections. The genomic data facilitated the identification of a novel viral lineage, TYLCV-Hasa, suggesting that the region is experiencing active local diversification alongside the introduction of distinct lineages from neighboring territories.

Our analysis provides critical insights into the evolutionary mechanisms driving this diversity. We observed that recombination is a pervasive force restricted to the DNA-A components, facilitating intra- and inter-species genetic exchanges, while DNA-B components remain evolutionarily conserved. Furthermore, population genetic analysis highlighted a distinct contrast between the stable, structured viral populations of Iran and the dynamic, expanding populations within the Arabian-Red Sea corridor, identifying the latter as a hotspot for viral admixture and variation. Notably, these infections were established in zucchini and snake gourd without the aid of DNA-Satellites, confirming these hosts as independent reservoirs for begomovirus evolution.

Collectively, these findings demonstrate that traditional primer-based diagnostics are insufficient for capturing the full extent of the localized viral diversity. The discovery of recombinant variants and the TYLCV-Hasa strain underscores the urgent need for continuous genomic surveillance. Implementing these advanced molecular tools will be essential for developing resistant crop varieties and IPM strategies, thereby supporting sustainable agriculture and food security initiatives in the region.

## Figures and Tables

**Figure 1 viruses-18-00075-f001:**
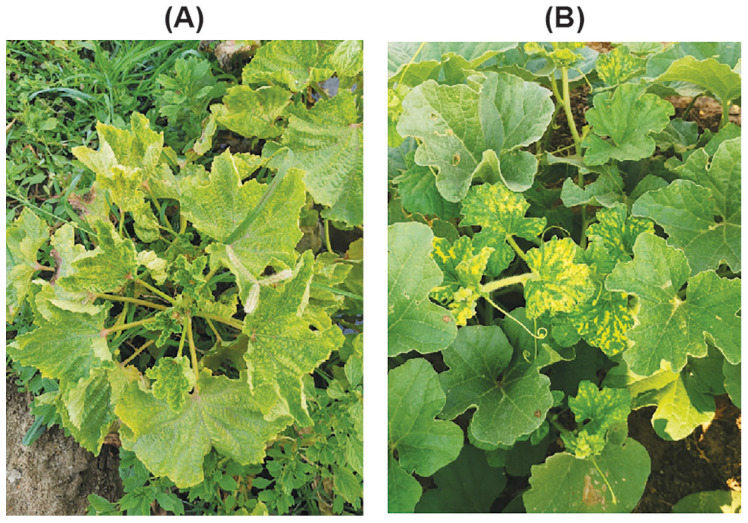
Virus-like symptoms observed on infected cucurbit plants. (**A**) Symptoms on zucchini (*Cucurbita pepo*) include yellow mosaic, vein banding and clearing, leaf rugosity, blistering, and cupping, as well as reduced leaf size and stunting. (**B**) Symptoms on snake gourd (*Trichosanthes cucumerina*) include bright interveinal yellow mosaic, puckering, and marginal curling, with the most severe deformation observed on young leaves.

**Figure 2 viruses-18-00075-f002:**
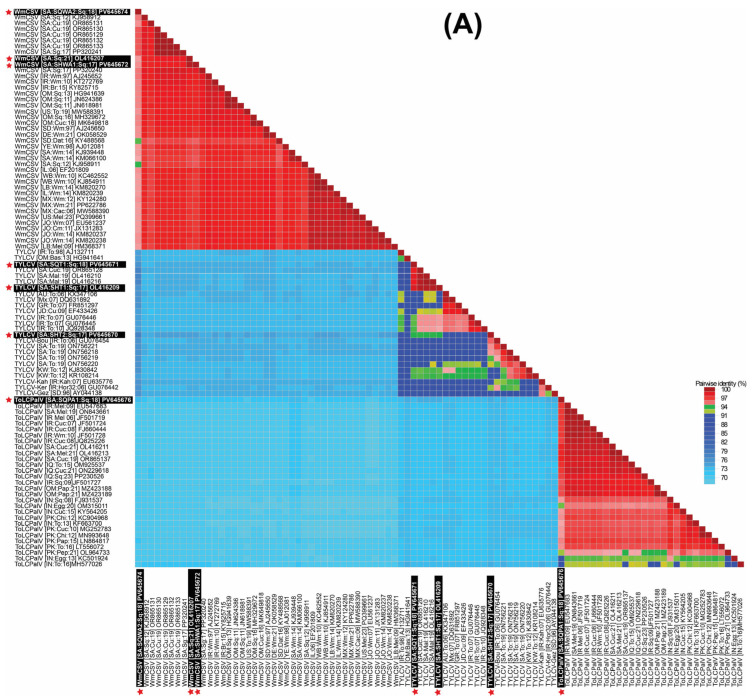
Sequence Demarcation Tool (SDT) heat map of pairwise nucleotide identity among complete begomovirus genomes. (**A**) DNA-A for WmCSV, ToLCPalV, and TYLCV. (**B**) DNA-B for WmCSV and ToLCPalV. Sequences were aligned with MUSCLE, and identities computed with SDT. The color scale spans 70–100%, with blue indicating lower identity and red indicating higher identity. The white text, boxed with a black background and red asterisks, marks the Saudi isolates identified in this study.

**Figure 3 viruses-18-00075-f003:**
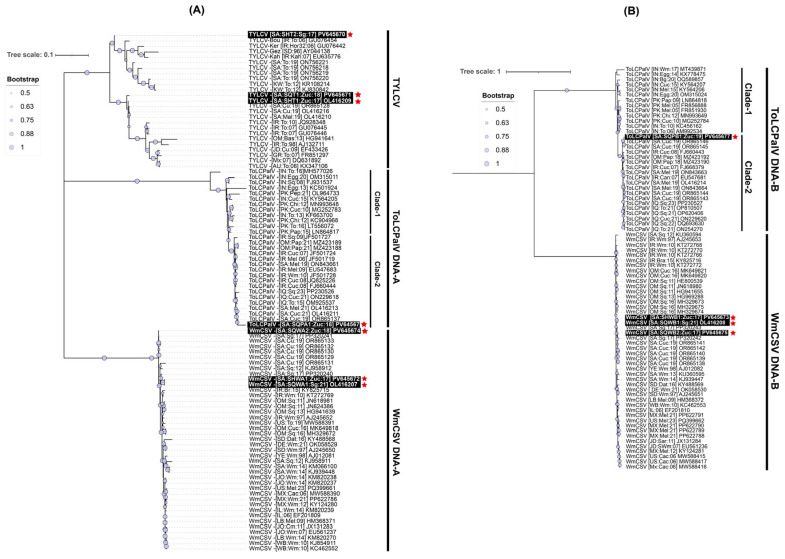
Phylogenetic analysis of begomovirus genomes. (**A**) DNA-A maximum-likelihood (ML) tree including TYLCV, ToLCPalV, and WmCSV. Two ToLCPalV DNA-A clades are indicated, and the Saudi TYLCV isolates group together within the TYLCV lineage. Support values are shown as filled circles, and the scale bar represents nt substitutions per site (0.1). (**B**) DNA-B ML tree including ToLCPalV and WmCSV. Two ToLCPalV DNA-B clades are indicated, and Saudi WmCSV genomes cluster within the regional WmCSV group. Support values are shown as filled circles, and the scale bar represents substitutions per site (1). Accession numbers are provided next to each taxon. The names of the sequences identified in this study are boxed in black and highlighted with red asterisks.

**Figure 4 viruses-18-00075-f004:**
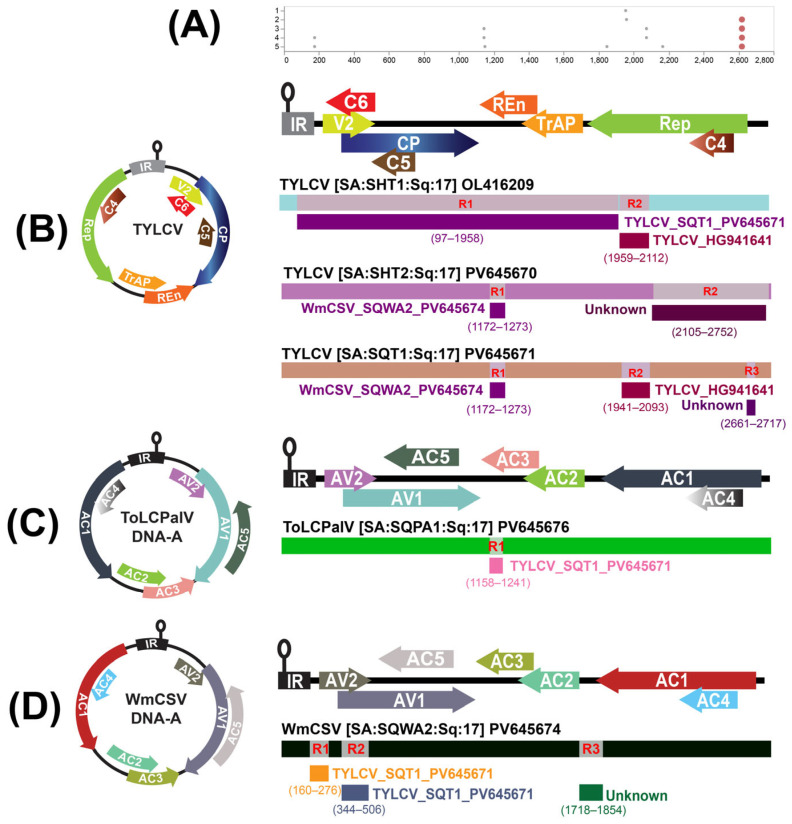
Recombination analyses of begomovirus DNA-A genomes. (**A**) GARD analysis of the DNA-A multiple-sequence alignment. The best-fit model identifies phylogenetically supported breakpoint regions along the genome; dots mark inferred breakpoints, and the complete genome illustration is shown for orientation. RDP-based recombination analysis of (**B**) TYLCV isolates SHT1, SHT2, and SQT1; (**C**) ToLCPalV DNA-A isolate SQPA1; and (**D**) WmCSV DNA-A isolate SQWA2. Circular and linear genomic illustrations are shown for the recombinant isolates with recombinant fragments, putative major/minor parents are indicated next to each fragment, and breakpoint intervals are given in parentheses. RDP suite methods (RDP, GENECONV, Bootscan, MaxChi, Chimaera, SiScan, and 3Seq) were used for panels, B-D and events were accepted when supported by ≥3 methods under Bonferroni-corrected *p* < 0.05. Colors denote coding regions and recombinant fragments as depicted.

**Figure 5 viruses-18-00075-f005:**
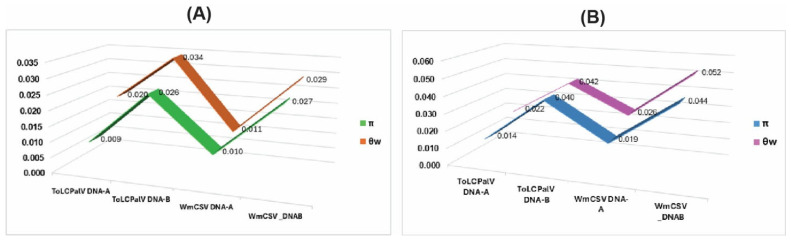
Comparative analysis of genetic diversity indices for ToLCPalV and WmCSV genomic components. (**A**) Genetic diversity profiles for the Iranian population (Pop-1), representing the historical diversification center. (**B**) Genetic diversity profiles for the Arabian-Red Sea population (Pop-2), representing the zone of viral expansion. The line graphs illustrate the values of nucleotide diversity (π) and Watterson’s estimator (θw) calculated for the DNA-A and DNA-B components of each virus. The divergence between θw and π indicates the presence of low-frequency variants and demographic expansion.

**Table 1 viruses-18-00075-t001:** Molecular detection and Illumina MiSeq whole-genome sequencing of begomoviruses in zucchini and snake gourd samples.

Plant	Sample	Place	TYLCV	WmCSV DNA-A	WmCSV DNA-B	ToLCPalV DNA-A	ToLCPalV DNA-B
Zucchini (*Cucurbita pepo* L.)	SqH1	Al-Ahsa	SHT1 (OL416209)	ND	ND	ND	ND
	SqH41	Al-Ahsa	ND	SHWA1 (PV645672)	SHWB1 (PV645673)	ND	ND
	SqSK7	Al-Ahsa	SQT1 (PV645671)	ND	ND	ND	ND
	SqA2	Qatif	ND	SQWA2 (PV645674)	SQWB2 (PV645675)	SQPA1 (PV645676)	SQPB1 (PV645677)
Snake Gourd (*Trichosanthes cucumerina* L.)	SqH33	Al-Ahsa	SHT2 (PV645670)	ND	ND	ND	ND
	SqA8my	Qatif	ND	SQWA1 (OL416207)	SQWB1 (OL416208)	ND	ND

Abbreviations: ND: Not Detected.

**Table 2 viruses-18-00075-t002:** Recombination events inferred for the identified begomovirus genomes using the RDP suite.

Recombinant	Event No.	Breakpoints *		Parents **		Methods ***	*p*-Value
		Start	End	Major	Minor		
TYLCV_SHT1 (OL416209)	R1	97	1958	TYLCV_GU076446 (95.3)	TYLCV_SQT1 (100.0)	***R***,*G*,***M***,***C***,***S***,***3S***	5.84 × 10^−28^
	R2	1959	2112	TYLCV_DQ631892 (95.0)	TYLCV_HG941641 (98.1)	***R***,***G***,***B***,***M***,***C***,***3S***	2.75 × 10^−12^
TYLCV_SQT1 (PV645671)	R1	1172	1273	TYLCV_EF4233426 (93.4)	WmCSV_SQWA2 (100.0)	***R***,***G***,*M*,*C*,***3S***	7.90 × 10^−11^
	R2	1941	2093	TYLCV_DQ631892 (94.1)	TYLCV_HG941641 (98.0)	***R***,***G***,***B***,***M***,***C***,***3S***	6.75 × 10^−12^
	R3	2661	2717	TYLCV_KJ830842 (94.7)	Unknown (TYLCV_ON756218)	***G***,***M***,***C***,***3S***	3.30 × 10^−12^
TYLCV_SHT2 (PV645670)	R1	1172	1273	TYLCV_EF433426 (92.8)	WmCSV_SQWA2 (100.0)	***R***,***G***,*M*,*C*,***3S***	7.96 × 10^−11^
	R2	2105	2752	TYLCV_SQT1 (100.0)	Unknown (TYLCV_JQ928348)	***R***,***G***,***B***,***M***,***C***,***S***,***3S***	1.15 × 10^−69^
ToLCPalV_SQPA1 (PV645676)	R1	1158	1241	Unknown (TYLCV_GU076446)	TYLCV_SQT1 (98.8)	***R***,***G***,*M*,*C*,***3S***	5.61 × 10^−12^
WmCSV_DNA-A_SQWA2 (PV645674)	R1	160	276	WmCSV_OK058529 (96.3)	TYLCV_SQT1 (100.0)	***R***,***G***,*M*,***C***,***S***,***3S***	6.28 × 10^−17^
	R2	344	506	WmCSV_KJ958912 (95.9)	TYLCV_SQT1 (99.4)	***R***,***G***,***B***,***M***,***C***,***S***,***3S***	3.35 × 10^−19^
	R3	1718	1854	WmCSV_PP320241 (97.3)	Unknown (TYLCV_OR865126)	***R***,***G***,***B***,*M*,*C*,***S***,***3S***	4.97 × 10^−16^

Abbreviations: 3S, Sequence Triplets; B, Bootscan; C, Chimarea; G, Geneconv; M, MaxChi; R, RDP; S, SiScan. * Each recombination breakpoint corresponds to the nucleotide position on the respective reference genome. ** Values in parentheses denote the percentage of nucleotide sequence identity between each recombinant isolate and the predominant begomovirus species identified as the major and minor parental sequences. *** Detection methods yielding *p*-values greater than 0.05 are indicated in bold italics, whereas the method producing the most significant *p*-value is underlined and presented in the respective column.

**Table 3 viruses-18-00075-t003:** Estimation of genetic diversity and neutrality test indices for TYLCV, ToLCPalV and WmCSV DNA-A and DNA-B in Pop-1 and Pop-2 datasets.

Virus Components	Number Seq	Polymorphic Sites (S)	Total Number of Mutations Eta (h)	InDel Sites	Number of Variants	Hd	π	k	h	θw	Neutrality Test	
											TD	FLD
Pop-1												
ToLCPalV DNA-A	24	204	220	2	35/8/0	1.00	0.009	25.75	24	0.020	−2.26	−3.47
ToLCPalV DNA-B	13	287	314	4	77/1/0	0.99	0.026	71.56	12	0.034	−1.35	−1.71
WmCSV DNA-A	7	76	77	2	19/1/0	1.00	0.010	26.29	7	0.011	−0.95	−0.94
WmCSV _DNAB	6	182	193	3	39/4/0	1.00	0.027	72.47	6	0.029	−0.93	−0.83
Pop-2												
ToLCPalV DNA-A	19	210	217	0	75/4/0	0.96	0.014	39.08	15	0.022	−1.55	−1.96
ToLCPalV DNA-B	16	379	413	5	253/32/0	0.93	0.040	108.77	11	0.042	−0.55	0.37
WmCSV DNA-A	23	262	281	3	112/16/0	0.98	0.019	53.26	18	0.026	−1.22	−1.56
WmCSV _DNAB	21	510	579	12	223/52/3	0.99	0.044	119.65	18	0.052	−1.06	−1.23

Abbreviations: Number seq—No of sequences used for the analysis; S—Number of polymorphic (segregating) site; Eta (h)—Total number of mutations; InDel sites—Insertion and deletions; Number of variants—Variable (polymorphic) sites (two/three/four variants); Hd—Haplotype (gene) diversity; π (Pi)—Nucleotide diversity (per site); k—Average number of nucleotide differences; h—Number of Haplotypes; θw—Watterson’s Theta; TD—Tajima’s D; FLD—Fu & Li’s D.

**Table 4 viruses-18-00075-t004:** Genetic differentiation estimates and Analysis of Molecular Variance (AMOVA) between Pop-1 and Pop-2 of ToLCPalV and WmCSV.

Virus Component	F_ST_ Value	*p*-Value/ Significance	K_ST_	S_nn_	AMOVA: % Variation (Among/Within Pops)
ToLCPalV DNA-A	0.178	<0.001	0.046	0.907	17.75%/82.25%
ToLCPalV DNA-B	0.106	<0.001	0.033	0.931	10.61%/89.39%
WmCSV DNA-A	0.118	0.006	0.028	0.967	11.79%/88.21%
WmCSV DNA-B	0.093	0.031	0.019	0.815	9.33%/90.67%

Abbreviations: F_ST_, pairwise fixation index; K_ST_, sequence-based differentiation statistic; S_nn_, nearest-neighbor statistic; AMOVA, analysis of molecular variance. *p*-values for F_ST_ were determined by 1000 permutations in Arlequin v3.5.

## Data Availability

All the complete genomic sequences identified from this study are available on NCBI GenBank database under the accession numbers PV645670-77 and OL416207-09.

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
