# Peer review of "High-Throughput Sequencing Reveals Regional Diversification of Cucurbit-Infecting Begomoviruses in Eastern Saudi Arabia"

_viruses, 2026, doi:10.3390/v18010075_

Round 1

Reviewer 1 Report

Comments and Suggestions for Authors

This study systematically characterized the genetic diversity of begomoviruses infecting cucurbit crops in eastern Saudi Arabia using high-throughput sequencing. From zucchini and snake gourd samples, we identified three viruses—Tomato yellow leaf curl virus (TYLCV), Watermelon chlorotic stunt virus (WmCSV), and Tomato leaf curl Palampur virus (ToLCPalV)—and discovered a locally evolved novel TYLCV strain (TYLCV-Hasa). The research revealed frequent interspecies genetic recombination among viruses, which was restricted to the DNA-A component. It detected no DNA satellite molecules, confirming that cucurbit crops can serve as independent viral reservoirs. These findings provide critical genomic evidence for molecular surveillance, early warning, and disease resistance breeding against begomoviruses in eastern Saudi Arabia. However, the present manuscript is deficient in several respects, including deficiencies in English language proficiency and inaccuracies in text formatting. Therefore, I recommend a Major revision of the manuscript. Some comments are given below.

  1. The quality of the English in the manuscript is a major concern. On first reviewing the paper, I made numerous edits/changes to the text and found myself unnecessarily distracted from assessing the paper on scientific grounds. There is no doubt that the quality of the paper is affected. Authors must proofread the full text carefully to correct possible grammatical errors and typos.
  2. The information in some tables and figures is not clear enough. It is recommended to adjust the clarity of the images.
  3. RCA is a rolling circle amplification method that may be biased toward high-abundance circular DNA; however, qPCR quantification was not employed to verify whether the proportions of viral components were distorted before and after RCA.
  4. The Iranian population (Pop-1) has far fewer samples than the Arabian population (Pop-2), resulting in distorted diversity comparisons. Sampling time differences were not considered (Iranian samples mostly from 2010–2015, Saudi samples from 2023–2025). Population differentiation statistics such as Fst and AMOVA were not performed; Tajima's D alone is insufficient to describe evolutionary.
  5. Methods 2.2 and Results 3.1 describe that six samples were each sequenced once, without assessing technical variation in NGS library preparation and sequencing, thus precluding discrimination between genuine viral variation and technical noise.

Author Response

Point-by-Point Response to Reviewer Comments

Reviewer 1:

Comment 1:

The quality of the English in the manuscript is a major concern... Authors must proofread the full text carefully to correct possible grammatical errors and typos.

Response:

We sincerely apologize for the grammatical errors and language inconsistencies in the original submission. We have taken this comment very seriously. The entire manuscript has been extensively edited and proofread by a native English speaker/professional editing service to improve flow, clarity, and grammatical accuracy. We have paid particular attention to sentence structure and technical terminology throughout the text.

Comment 2:

The information in some tables and figures is not clear enough. It is recommended to adjust the clarity of the images.

Response:

Thank you for pointing this out. We have revised all figures to ensure they meet high-resolution standards (300 dpi). specifically:

  • Figure 1: We have adjusted the contrast and brightness to clearly show the viral symptoms.
  • Figure 2 (SDT Heatmap): We have increased the font size of the axis labels and accession numbers to ensure legibility.
  • Figure 3 (Phylogenetic Trees): The trees have been re-rendered to ensure branch support values and taxon names are clearly readable without zooming.
  • Tables: We have reformatted the tables to ensure proper alignment and clearer separation of columns for better readability.

Comment 3:

RCA is a rolling circle amplification method that may be biased toward high-abundance circular DNA; however, qPCR quantification was not employed to verify whether the proportions of viral components were distorted before and after RCA.

Response:

We acknowledge the reviewer’s valid point regarding the preferential amplification of small circular DNA by Phi29 polymerase, which can indeed alter the relative proportions of viral components compared to the original sample.

However, we would like to clarify that the primary objective of this study was viral discovery, identification, and full-genome assembly, rather than absolute quantification of viral load. RCA was utilized specifically because of its ability to enrich circular begomovirus genomes from low-titre plant samples to facilitate successful NGS assembly. To ensure the reliability of the presence of these viruses, we validated the NGS findings using standard PCR with component-specific primers followed by Sanger sequencing (as detailed in Section 2.4 of the Materials and Methods). The PCR validation confirmed the presence of the specific viruses identified via RCA-NGS. We have added a statement in the Discussion section acknowledging the quantitative limitations of RCA while highlighting its strength in qualitative viral discovery (Lines 585-588).

Comment 4:

The Iranian population (Pop-1) has far fewer samples than the Arabian population (Pop-2), resulting in distorted diversity comparisons. Sampling time differences were not considered... Population differentiation statistics such as Fst and AMOVA were not performed; Tajima's D alone is insufficient.

Response:

We appreciate this critical insight regarding the population genetics analysis.

  1. Sample Size & Time: We acknowledge the disparity in sample sizes and collection dates. This is a common limitation when comparing novel field data with historical GenBank deposits. To address this, we have added a caveat in the Materials and Methods (Lines 216-219) and Results (Lines 428-429) sections explicitly stating that the Pop-1 dataset represents a historical "snapshot" compared to the contemporary Pop-2 dataset.
  2. Statistical Analysis: We agree that Tajima's D is insufficient for defining population differentiation. In the revised manuscript, we have performed Pairwise Fixation Index (FST) and Analysis of Molecular Variance (AMOVA) calculations. These analyses provide a statistical basis for the genetic differentiation between the Iranian and Arabian populations, strengthening our conclusion regarding the distinct evolutionary trajectories of these groups. This new data has been added as a new Table 4 and discussed in the text (Lines 240-247, 444-465, 550-563).

Comment 5:

Methods 2.2 and Results 3.1 describe that six samples were each sequenced once, without assessing technical variation in NGS library preparation and sequencing, thus precluding discrimination between genuine viral variation and technical noise.

Response:

We agree with the reviewer that technical replicates are essential when the goal is to analyze intra-host viral diversity (quasispecies) or rare single-nucleotide variants (SNVs). However, the scope of this study was to assemble consensus genomes to identify the species and strains circulating in the region.

  • For consensus assembly, the high depth of coverage achieved by the Illumina MiSeq platform (ranging from ~2.2 to 2.7 million reads per sample) provides high confidence in the base calling of the dominant viral genome sequence, minimizing the impact of technical noise on species identification.
  • We applied strict quality filtering (Trimmomatic) and depth thresholds (>20x) during assembly to ensure the reliability of the consensus sequences.
  • We have revised the Discussion to clarify that our analysis focuses on the consensus genome sequences for taxonomic and phylogenetic placement, rather than intra-host variant clouds, where technical replicates would be mandatory (Lines 578-582).

Reviewer 2 Report

Comments and Suggestions for Authors

Authors Muhammad Naeem Sattar and colleagues submitted a manuscript entitled „High-throughput Sequencing Reveals Regional Diversification of Cucurbit-Infecting Begomoviruses in Eastern Saudi Arabia“.

The authors focused on research into begomoviruses infecting cucurbit vegetables in Saudi Arabia.

The survey focused on both field and greenhouse crops of cucurbit plants in two regions of the country. Symptomatic samples were tested for the presence of begomoviruses using PCR, RCA, and HTS.

Virus identification was based on the sequences obtained. Three viruses predominated in symptomatic plants: TYLCV, WmCSV, and ToLCPalV. The authors identified a new TYLCV-Hasa strain. Mixed infections were common. The authors performed phylogenetic studies and recombination analysis on the viruses detected.

The manuscript is thoroughly written. The introduction provides a comprehensive summary of the issue of begomoviruses. The materials and methods described are very detailed and thorough. The results are described in great detail and properly documented. The discussion compares the findings in this work with previously published results.

The results achieved expand knowledge in the field of begomovirus diversity. They are also important for agricultural production in Saudi Arabia and in countries with a similar climate and similar crop composition.

I recommend expanding the discussion with a paragraph devoted to other groups of viruses that infect cucurbit vegetables in Saudi Arabia.

Author Response

Reviewer 2:

Comment 1: "I recommend expanding the discussion with a paragraph devoted to other groups of viruses that infect cucurbit vegetables in Saudi Arabia."

Response:

We thank the reviewer for their positive assessment of our work and for this valuable suggestion. We agree that placing our begomovirus findings within the broader context of cucurbit virology in the region strengthens the manuscript. We have added a new paragraph in the Discussion section (Lines 481-487) that summarizes the prevalence of other major cucurbit-infecting viruses in Saudi Arabia, including Zucchini yellow mosaic virus (ZYMV), Watermelon mosaic virus (WMV), Cucumber mosaic virus (CMV), and Cucurbit yellow stunting disorder virus (CYSDV). We have cited relevant local studies to support this addition.

Round 2

Reviewer 1 Report

Comments and Suggestions for Authors

I recommend the acceptance of the revised manuscript for publication.